# Parasitic Characteristics of Ticks (Acari: Ixodidae) Collected from Water Deer (*Hydropotes inermis argyropus*) and Spatiotemporal Distribution Prediction within Host-Influenced Cultivated Areas

**DOI:** 10.3390/ani14081153

**Published:** 2024-04-10

**Authors:** Kiyoon Kim, Kyungmin Kim, Kwangbae Yoon, Yungchul Park

**Affiliations:** 1College of Veterinary Medicine, Chungbuk National University, Cheongju 28644, Republic of Korea; anikest@naver.com; 2Interdisciplinary Program of Eco Creative, Ewha Womans University, Seoul 03760, Republic of Korea; kyungminkim0829@gmail.com; 3Research Center for Endangered Species, National Institute of Ecology, Yeongyang 36531, Republic of Korea; rhkdqoek@nie.re.kr; 4Division of Forest Science, Kangwon National University, Chuncheon 24341, Republic of Korea

**Keywords:** cultivated area, distribution, *Haemaphysalis longicornis*, *Hydropotes inermis argyropus*, host–vector interaction, tick

## Abstract

**Simple Summary:**

The dispersion of ixodid ticks depends on their hosts, with tick density correlating positively with host density. Water deer, designated as harmful wildlife due to their foraging activities in croplands, are considered significant hosts for dispersing ixodid ticks within human residential areas. Over the course of two years, a substantial number of water deer captured while entering croplands were subjected to analysis for the presence of parasitic ixodid ticks. The findings revealed a notable mean intensity of nymphs and adults during summer, whereas larvae were found to be more abundant during fall. The number of larvae correlated positively with the population density of water deer. MaxEnt modeling predicted broad distribution areas for water deer in summer, coinciding with abundant crops, and similarly for nymphs and adults, which peak in summer. Forest valleys converted into croplands are expected to promote ixodid tick dispersion due to alignment with the habitat preferences of water deer and increased crop utilization. Since agricultural factors contribute significantly to the occurrence of tick-borne diseases, preventive measures such as selective land clearing and crop selection should be implemented to mitigate human contact with ixodid ticks in farming environments.

**Abstract:**

Tick-borne diseases result from human–ixodid tick encounters, making it crucial to understand host–ixodid tick interactions and host-dependent distribution for epidemiology and prevention. This study examines water deer patterns and ixodid tick interactions in designated croplands of South Korea over two years, finding that the highest deer presence occurred in July and the lowest in May, during crop harvesting. Four tick species were identified, with *Haemaphysalis longicornis* being predominant (92.2%). Tick life stage analysis revealed peak nymphs and adults in July and larvae in October. Nymph abundance correlated positively with nearby water deer. MaxEnt biodiversity prediction results indicated wider water deer distribution in summer, reflecting their tendency to use multiple croplands. Areas with nymphs and adults aligned with predicted deer presence in summer, while larval areas aligned in autumn. Increased agroforestry expanded water deer habitats, enhancing tick dispersion. Prevention involved minimizing human–deer encounters by strategic land use in tick-prone areas. This comprehensive study provides insights into preventing severe fever with thrombocytopenia syndrome in agricultural workers, emphasizing the need for targeted interventions based on host behavior and tick life stages in different seasons.

## 1. Introduction

Human population expansion has led to human residential area increases, transforming vertebrate-inhabited regions into croplands [1]. Natural habitat loss has compelled wild animals to cohabit with humans, resulting in overlapping habitats that intensify human–wildlife conflicts, such as the utilization of food sources by wild animals within croplands [2,3] and the resurgence and spread of zoonotic diseases [4].

South Korea recognizes various zoonotic diseases, among which are tick-borne illnesses caused by ixodid ticks, encompassing severe fever with thrombocytopenia syndrome (SFTS), Lyme disease, and tick-borne viral encephalitis [5]. Ixodid ticks target a broad range of mammals, including humans, and transfer pathogens during blood feeding. Ixodid tick abundance correlates with host density. Habitat humidity plays a crucial role in tick survival during waiting periods but they meet their water requirements through the blood of the host during feeding [6,7,8]. Increased host density in a specific area of ixodid tick dispersion enhances blood-feeding opportunities, reducing exposure to adverse environmental factors and allowing for a high tick density via favorable effects on tick survival and developmental stages [9,10,11]. Consequently, an area with a high host density can be considered a habitat with a high ixodid tick abundance.

Ixodid ticks migrate in search of hosts, with migration ranges typically localized at approximately 8 m [12,13]. The length of the parasitic period increases with the migration distance of the host [14]. Upon detachment from the host after blood feeding, the dispersion range of the tick may fall within the home range of the host [15,16,17,18]. Therefore, the home range of the host can be interpreted as the habitat and dispersion range of ixodid ticks. Considering host–ixodid tick interactions, contact between humans and ixodid ticks is postulated to occur based on the host species inhabiting the vicinity of human residential areas at a high density and the characteristics of how the host utilizes the habitat.

Water deer (*Hydropotes inermis* Swinho, 1870) comprise two subspecies: *H. inermis inermis* in the eastern region of China and *H. inermis argyropus* across South Korea, listed as vulnerable on the International Union for Conservation of Nature (IUCN) Red List [19]. In China, *H. inermis inermis* faces declining numbers due to habitat loss and fragmentation, whereas *H. inermis argyropus* thrives throughout South Korea, where its ecosystem lacks natural predators [20,21,22,23]. A generalist species, water deer adeptly inhabit croplands, wetlands, grasslands, and forests [23], displaying edge species characteristics [19]. Notably, their foraging activity in cultivated areas causes damage to croplands, leading to their designation as harmful wildlife since 2005. The annual capture of water deer steadily increased from 2014 to 2018 to a total of 651,852 individuals, defining the species as the most abundantly captured harmful wildlife in South Korea [24]. A study on animal roadkill further highlighted water deer as the dominant species (44.84%) among wild land animals killed in South Korea [25], emphasizing frequent habitat overlaps with humans. Therefore, among land mammalian species acting as hosts for ixodid ticks, water deer are presumed to play a crucial role in the potential route of contact between humans and ixodid ticks.

In tick-borne diseases, SFTS exhibits a consistent infection rate, particularly affecting older adults ≥ 50 years, predominantly comprising agricultural workers [26,27,28]. Prior studies on parasitic ixodid ticks found in water deer (a host species frequently accessing croplands) primarily focused on rescued deer or carcasses from roadkill [29,30,31,32]. However, live animal examinations could considerably enhance their dispersion predictions. Estimating the abundance of host parasitic ixodid ticks using roadkill carcasses proves challenging as ticks detach from host carcasses. Visual examination limitations in defining the tick index arise due to the impossibility of capturing all wandering ticks on the host. Additionally, determining the annual mean number of parasitic individuals and their life stage is challenging, as most studies on ixodid ticks using hosts concentrate on distribution within specific administrative areas or pathogen detection [33,34,35].

This study collected parasitic ixodid ticks from water deer carcasses in croplands over two years. We aimed to elucidate previously undisclosed cultivation area utilization patterns and calculate the parasitic tick index of water deer. Additionally, we analyzed the spatiotemporal occurrence patterns of ticks in cultivated areas influenced by their hosts. This research contributes to the examination of the dispersal role of water deer, designated as harmful wildlife in areas near human habitats. Furthermore, this study is anticipated to enhance our comprehension of the spatiotemporal patterns of ticks dispersing near cultivated areas. Via analysis of the interactions among vectors, hosts, and habitats in disease transmission, these findings are expected to make valuable contributions to the prevention of tick contact among high-risk populations, such as farmers, and the elucidation of epidemiological relationships associated with tick-borne diseases.

## 2. Materials and Methods

### 2.1. Study Area

This study took place in the croplands of Inje-gun, Gangwon-do, South Korea. According to the land cover map classification, Inje-gun is predominantly covered by forests, constituting 91.38% of the total area of 1644.97 km^2^ (Figure 1). Additional land cover includes grasslands (2.84%), croplands (4.93%), water bodies (0.76%), and urban areas (0.1%). The study site and location where deer were captured fell within a designated region under the supervision of an official who issued a permit for capturing water deer deemed as harmful wildlife. This official accompanied the investigators throughout the study period. The main crops cultivated in Inje-gun are corn, rice, soybean, potato, winter cabbage, and white radish (Division of Environmental Protection, Inje-gun County Office).

### 2.2. Capturing Water Deer and Detection Site Coordinates

The capture and monitoring of water deer were conducted between 10 May and 10 November 2019, and between 19 May and 12 November 2020, during the legally permitted annual activity periods for capturing harmful wildlife. The survey was carried out three times (three nights and four days) in May and October, twice in June, and twice (two nights and three days) in July, August, and September. The total number of days dedicated to capturing activities was 36, consistent for both 2019 and 2020. Capture and monitoring primarily occurred along designated routes within the permitted area covering a two-lane road spanning 79.57 km (Figure 1). Wildlife access in croplands near the road was inspected during peak hours for wild animal appearances from 21:00 to 04:00 on the following day while driving at speeds of 10 to 20 km/h. Using searchlights to detect reflected light from the eyes of animals and thermal imaging from an infrared camera installed in the vehicle, the researchers identified wildlife access and determined the species of harmful wildlife. Coordinates were recorded for confirmed access. Gun-capturing was employed for confirmed wildlife access, with carcasses stored in sealed plastic bags and farming sacks to prevent tick escape. In instances where gun-capturing was impractical or too close to residential areas, only the coordinates of the respective sites were recorded.

### 2.3. Collection and Identification of Water Deer and Parasitic Ixodid Ticks

To optimize ixodid tick collection from water deer carcasses, the plastic bag containing the carcass was opened and the carcass was transferred to a large plastic basin before resealing the bag. Furthermore, the skin was carefully removed from the tail end by making incisions from the four ankles to the dorsal and lower snout parts. Ticks on the skin surface and those that migrated to the muscle surface were meticulously collected. Any remaining ticks in the plastic bag were collected, and the skin was submerged in water to collect the ticks as they drowned or floated (Figure 2).

All collected ticks were stored in a dedicated tick collection tube to prevent any potential escape. The collected ticks underwent a thorough rinsing in 70% alcohol to eliminate any traces of host blood and hair, thus aiding in species identification through morphological characteristics. Furthermore, to prevent the detachment of ticks during the identification process, they were carefully stiffened on ice packs, ensuring the safety of researchers. The identification of ixodid ticks obtained from water deer relied on morphological observations conducted using a dissecting microscope (Olympus, Tokyo City, Japan, SZ61-ST) [36]. Additionally, life stages (larva, nymph, and adult), and, in the case of adult ticks, sex, were systematically classified. Larvae that resisted species identification via morphological analyses were categorized at the genus level. All individuals, once identified, were segregated by category and stored at −70 °C. 

### 2.4. Statistical Analysis

To assess the level of ixodid tick infection in the host, the mean intensity, presented as the mean ± standard error, was employed to depict the infection level by month and tick life stage [37]. Furthermore, the Kruskal–Wallis test was conducted to identify significant differences in tick appearances at each life stage by month (*p* < 0.05). The Mann–Whitney post hoc test was then employed to scrutinize monthly variations in appearance (*p* < 0.005).

### 2.5. Cropland Use Characterization by Water Deer and Parasitic Ixodid Tick Distribution Prediction

To forecast the areas of potential human–water deer conflicts, a habitat suitability test for water deer was conducted using coordinates from sites where they were detected, sourced from the Global Biodiversity Information Facility (GBIF; GBIF.org, accessed in 2022). The essential variables in the habitat suitability test included environmental data at a 0.001 resolution and microclimate (Worldclim.org ver. 2.1) [38], land type, and topography data (Table 1). Duplicate coordinates were eliminated to mitigate spatial autocorrelation stemming from sampling bias. The model underwent training via five repeated tests using the cross-validate run type. The mean of these repeated tests constituted the final model, with the criterion of percent contribution (>5) and permutation importance (>5) applied to the variables within the initial complete model. Model fit was presented using the area under the receiver operating curve (AUC) and true scale statistic (TSS) determined using R software version 4.3.2. [39]. A model fit was considered satisfactory based on the criteria of mean AUC ≥ 0.6 and mean TSS ≥ 0.4. The water deer habitat suitability model generated was layered with sighting coordinates to predict human–water deer conflict zones. The coordinates of water deer, acting as ixodid tick hosts, were used to predict the distribution of ixodid ticks as dispersed by water deer. The previously described method (forecasting the areas of potential human–water deer conflicts) was employed with weightings based on ixodid tick numbers.

## 3. Results

### 3.1. Capturing Water Deer

Over the course of 66 field investigations conducted from May to November in 2019 and 2020, a total of 48 water deer were captured among 152 observed water deer. The highest annual deer count was observed in July for both years, with *n* = 16 in 2019 and *n* = 30 in 2020. Conversely, the lowest count was recorded in May, with *n* = 8 in 2019 and *n* = 1 in 2020 (Figure 3). In 2020, the highest number of captured deer occurred in August, reaching *n* = 16 (Figure 4), whereas the lowest count was *n* = 1 in May. In August 2019, adverse weather conditions, that is, prolonged rainfall from 15 July to 31 August, and restrictions on firearm use due to a continuous stream of visitors in the region led to the lowest count of captured deer at *n* = 1, limiting the scope of investigations during that period.

### 3.2. Water Deer and Parasitic Ixodid Tick Communities

Ixodid ticks were collected from 39 of the 48 water deer captured from May to November. Excluding larva-stage ticks identified at the genus level, the total number of ixodid ticks for nymph and adult stages amounted to 6620. Four species of ixodid ticks utilizing water deer as hosts were identified: *Haemaphysalis longicornis* Neumann, 1901, *Haemaphysalis flava* Neumann, 1897, *Haemaphysalis japonica* Warburton, 1908, and *Ixodes nipponensis* Kitaoka & Saito, 1967 (Table 2). 

The predominant species, *H. longicornis*, displayed the highest count at 6101 (92.2%), followed by *H. flava* at 385 (5.8%), *I. nipponensis* at 79 (1.2%), and *H. japonica* at 55 (0.8%). The capture count of *H. longicornis* peaked in July and declined from August onwards, with the count dropping below that of *H. flava* and *I. nipponensis* in October. During the summer cropland period (June–August), coinciding with water deer appearance, the appearance rate and count of ixodid species, except for *H. longicornis*, were low. Conversely, in the autumn months (September–November), as *H. longicornis* numbers decreased, the appearance rate and count of other tick species surpassed those of *H. longicornis* (Table 3).

No ticks were found on seven deer captured during the August rainfall period and two water deer captured in both November and May. Additionally, one water deer captured in May was excluded from statistical analysis due to a small sample size. A total of 21,976 ixodid ticks were collected, averaging 563.48 ± 74.67 per deer. The larva-stage, nymph-stage, and adult-stage ticks accounted for 69.9% (15,353), mean: 451.56 ± 92.90; 22.5% (4956), mean: 127.07 ± 18.54; and 7.6% (1667), mean: 42.74 ± 7.48, respectively (Table 3). Larva-stage ticks were significantly more abundant in October, whereas adult-stage ticks peaked in July (*p* < 0.005; Table 2). Larva-stage ticks were first observed in June, peaking in October; nymph-stage ticks were initially seen in May, peaking in July; and adult-stage ticks followed a similar pattern, first appearing in May, peaking in July, and then decreasing until October (Figure 4).

Examining the correlation between water deer numbers near croplands and ixodid tick counts in each life stage, a positive correlation was observed throughout all seasons at the nymph stage, despite the higher numbers in summer. The adult-stage ixodid tick count exhibited no correlation with water deer numbers but was correlated with the nymph-stage ixodid tick count (Table 4).

### 3.3. Prediction of Water Deer Distribution

The MaxEnt prediction model for human–water deer conflict zones in summer yielded a mean AUC of 0.8409 ± 0.0287 and a mean TSS of 0.5802 ± 0.1449. The highest permutation importance was for the distance to evergreen forest, followed by the distance to agricultural paddy, water deer habitat suitability, distance to shrubs, and distance to urban areas (Figure 5). In autumn, the prediction model showed a mean AUC of 0.9683 ± 0.0111 and a mean TSS of 0.8112 ± 0.0794. The highest permutation importance was for the distance to evergreen forests, followed by distances to agricultural lands, agricultural paddies, shrubs, and Bio7 (annual temperature range) (Figure 5).

The areas predicted for water deer appearance near croplands in summer corresponded to the distribution of agricultural lands in Inje-gun, with high predicted appearances across all areas except forested regions. In contrast, the predicted areas for water deer appearance in autumn were limited to a very narrow scope (Figure 5).

Among the areas predicted to highlight the distribution of water deer, those with a probability ≥0.6 were 167.97 km^2^ in summer and 37.74 km^2^ in autumn, indicating larger areas in summer. When overlaying these areas on the human residential zone (85.28 km^2^) in the classified land cover map encompassing nine variables, such as housing areas, urban areas, croplands, and orchards, the overlapping areas were 17.11 km^2^ in summer and 16.44 km^2^ in autumn. This result indicates that 10.19% of the areas predicted to exhibit widespread water deer distribution in summer and 43.55% of the areas predicted to exhibit limited water deer distribution in autumn overlap with human residential areas (Figure 6).

### 3.4. Distribution Prediction of Parasitic Ixodid Ticks

The distribution of ixodid ticks in each life stage was predicted using MaxEnt software Version 3.4.4 (New York City, NY, USA) and applying weight values assigned by the number of individuals based on water deer coordinates (Figure 7). The prediction model for adult-stage ixodid ticks dispersed by water deer showed a mean AUC of 0.8048 ± 0.0403 and a mean TSS of 0.6035 ± 0.1956. Valid variables included distances to agricultural lands in general, shrubs, deciduous forests, and Bio13 (precipitation of wettest month). Among these, the distance to agricultural lands in general displayed the highest contribution (69.4) and permutation importance (58).

The prediction model for nymph-stage ticks showed a mean AUC of 0.8713 ± 0.0034 and a mean TSS of 0.7366 ± 0.024. Valid variables were the distance to agricultural lands in general, water deer habitat suitability index, distances to agricultural paddies, shrubs, wetlands, and Bio13 (precipitation of wettest month). The distance to agricultural lands in general had the highest contribution (25.6), while water deer habitat suitability had the highest permutation importance (28.9).

The prediction model for larva-stage ticks showed a mean AUC of 0.8748 ± 0.0024 and a mean TSS of 0.7547 ± 0.0563. Valid variables were the distances to agricultural lands in general, wetlands, urban areas, and water deer habitat suitability index. The distance to agricultural lands in general (56.2) and the distance to shrubs (41.6) had the highest percent contribution and permutation importance, respectively (Figure 8).

The highest dispersion by water deer was found in adult-stage ticks, overlapping with most cropland areas except high-altitude or small-scale forest croplands. For ixodid ticks in all life stages, the importance of distance to agricultural lands in general was high, predicting tick dispersion in the vicinity of croplands and residential areas. Ixodid ticks in the nymph and adult stages were abundant in summer but were not collected during the rainfall period. The probability of ixodid tick appearance decreased with increasing Bio13 (wettest month precipitation). The distribution of nymph-stage and adult-stage ticks was wide, consistent with the water deer distribution prediction map of summer, whereas the distribution of larva-stage ticks was narrow, consistent with the water deer distribution prediction map of autumn.

When areas with a distribution probability ≥0.6 were analyzed according to the ixodid tick life stage, the distribution area was 198.50, 33.32, and 15.47 km^2^ for the adult, nymph, and larva stages, respectively. The area overlapping with residential areas was 74.95 km^2^ (adult stage), 7.03 km^2^ (nymph stage), and 6.00 km^2^ (larva stage), predicting distributions of 87.89, 8.24, and 7.04%, respectively, across cropland areas (85.28 km^2^).

## 4. Discussion

Water deer inhabit a variety of areas nationwide, ranging from forests to mountain edges and vegetative colonies near water. They are known to prefer water and mountain edges [20,23]. Small-scale croplands resulting from forest reclamation create forest edges (ecotones), which are preferred water deer habitats. The density of lowland water deer exhibiting such a preference increases in the vicinity of reclamation areas. Lowland water deer have a more limited variety of food sources compared to highland forest water deer. The density of lowland water deer is higher than that of highland water deer (6.73 vs. 1.91 individuals/km^2^) [20], possibly attributed to croplands providing stable food supply areas and variations in vegetation structure and diversity between lowland and forest areas.

Water deer with access to croplands do not need to roam for food, leading to them inhabiting a narrow range at a high density compared to forest-dwelling water deer. Additionally, water deer migrate along water, using water plant communities near croplands as hiding places [40]. Farming irrigation canals and tributaries distributed in lowland valleys near croplands satisfy the habitat preference requirements of water deer—water, food sources, and hiding places. Ultimately, forest reclamation areas in lowland regions connected to forest valleys in Inje-gun can be viewed as preferred habitats for water deer, satisfying all three essential factors for wildlife habitation (Figure 9).

Water deer seldom consume tree seeds or fruits and prefer plants of the Cyperaceae family and herbaceous plants. They also feed on crops such as chili, white radish, winter cabbage, corn, potato, pumpkin, rice, soybean, and sweet potato [20,40,41]. In Inje-gun, the total reported cases of cropland damage from 2019 to 2020 were 255, with 81.6% of the damage across 35 crops attributed to five main crops: corn (57.3%), potato (7.8%), chili (6.7%), soybean (6.3%), and rice (3.5%) (Division of Environmental Protection, Inje-gun County Office). The substantial cropland access by water deer is closely associated with the damage to three main crops: corn, soybean, and chili. It is presumed that water deer begin appearing in croplands from April or May during crop seeding, peaking in July before the harvest of the three main crops, and decreasing in August with the rapid decline in crops due to harvesting. Following this, water deer resume accessing croplands from September onwards, targeting autumn-harvested crops such as rice, white radish, and winter cabbage. By November, with the completion of all cropland harvests, the use of cropland food sources by water deer ceases entirely.

There was a significant positive correlation between the number of water deer and nymph-stage ixodid ticks during the summer. Additionally, the number of nymph- and adult-stage ticks showed a positive correlation. This is likely due to the increased number of water deer acting as local hosts, increasing the likelihood of encounters between water deer and nymph-stage ticks. It may also be attributed to the increased number of nymphs maturing into adults—the next life stage of ticks—after completing blood feeding. Therefore, lowland water deer inhabiting areas with a high density overlapping with human residential areas can be considered the primary host influencing the abundance of ixodid ticks.

The parasitic ixodid ticks associated with water deer were notably absent during the rainy period in August. The distribution prediction for nymph- and adult-stage ticks, abundant in summer, indicated an appearance probability decrease with an increase in Bio13 (wettest month precipitation). Since ticks can progress to the next life stage only after successful blood feeding, the most influential factor in the life cycle of ixodid ticks is presumed to be the rainy period in water deer habitats, their primary host in lowland areas.

Water deer habitats with high rates of rainfall infiltration are likely to impact the host search period and pattern of ixodid ticks, which generally await a chance encounter with the host at resting areas or migration routes [42]. This mechanism depends on continuously released carbon dioxide by the host in resting areas and the secretions on the host surface during migration or food-related activities [42,43]. Prolonged rainfall infiltration is predicted to have the following effects on the host search pattern of ixodid ticks, in addition to host parasitic behaviors: (1) the host may leave the habitat resting area and stay in an area without infiltration during prolonged rainfall, preventing ticks from detecting their hosts, and (2) the search for the host by ixodid ticks becomes challenging because the outer secretions in previous host activity areas could be washed away by rainfall infiltration.

In the rainy season (July–August), the number of observed or captured water deer increased, suggesting negligible effects of prolonged rainfall on host behavior. However, the absence of parasitic ixodid ticks on water deer captured during this period may imply a decrease in the dispersion of ixodid ticks in the overlapped areas of water deer inhabitation and human residence. This finding is expected to be valuable evidence for developing measures in the context of preventive and epidemiological investigations of ixodid ticks, emphasizing the need for further studies.

The life cycle of ixodid ticks encompasses egg, larva, nymph, and adult stages, with each stage spending a specific duration in the external environment, detached from their animal hosts. Tick survival and reproduction are intricately linked to external environmental factors such as temperature, humidity, and vegetation. For example, ixodid tick eggs do not hatch at temperatures ≤12 °C [44], with a significantly low hatching rate in high-temperature or high-relative-humidity conditions [45,46]. During the cold winter months, ixodid ticks enter a dormant phase, while their period of activity extends from April to November.

The pattern of ixodid tick appearance in each life stage can be elucidated as follows: (1) Nymphs: The larvae increase in abundance in September–October, after which they succeed in blood feeding and then overwinter under a vegetation mat layer. Following metamorphosis in the subsequent spring, their activity as nymphs resumes in April–May. In June–July, as water deer’s access to croplands increases, enhancing the chance of encounter between ticks and their water deer hosts, the rate of successful blood feeding rises. (2) Adults: After approximately 26 days of metamorphosis, the nymphs begin to appear in June, reaching peak abundance in July. Subsequently, these nymphs, showing the highest rate of successful blood feeding due to increased encounters with hosts, metamorphose into adults in August. However, prolonged rainfall from the end of July to mid-August hinders host encounters, leading to a decrease in the number of parasitic ticks on the hosts. (3) Larvae: From the eggs laid by the adults that successfully completed blood feeding and breeding in June, the larvae start to appear at the end of July. Eggs laid by adults that completed blood feeding in July and August are buried under the vegetation mat layer, but due to increased humidity during continuous rainfall, these eggs cannot hatch. By the end of the rainy season and with stabilized humidity in September, the eggs finally hatch, and the larvae start to appear and feed on hosts accessing croplands in autumn. The larvae then spend the winter under the vegetation mat layer and resume activity as nymphs around April of the following year.

*Haemaphysalis longicornis* was consistently dominant in the study period, comprising the highest proportion (92.2%) among all identified ixodid ticks. This species primarily utilizes water deer as hosts, showing a peak in appearance during summer and a decline in autumn. Given the prediction that the utilization of agricultural land by the host, water deer, would be more extensive during the summer, *H. longicornis*, which records a higher population during the summer, is expected to be more widely dispersed within these areas due to the utilization characteristics of water deer. Consequently, there is an increased likelihood of the virus carried by *H. longicornis*, known as the primary vector of domestic SFTS [47,48], being transmitted to rural farmers within agricultural lands. Therefore, it is speculated that this could be associated with a high seroprevalence rate among rural farmers in domestic settings, along with high incidence factors related to agriculture.

Ixodid ticks collected from reptiles in South Korea between April and November belonged to the *Ixodes* and *Amblyomma* genera, with the *Haemaphysalis* genus—here, the dominant species—being notably absent [49]. Consequently, the prevalence of *H. longicornis* in water deer is likely due to host preference variations among different ixodid tick species. Moreover, when comparing tick infestations between water deer and roe deer captured in the same area at the same time [50], it was noted that ticks infesting roe deer were more abundant. This indicates potential differences in host preference among closely related host species. However, it is crucial to acknowledge that this study focused exclusively on water deer as the host species, limiting the exploration of density and preferred habitats across other potential host species. Future research should encompass diverse host species found in areas adjacent to human habitats to comprehensively understand respective densities and preferred habitats.

The MaxEnt model predicted a broader distribution of water deer accessing croplands in summer (167.97 km^2^) compared to autumn (37.74 km^2^). The model for summer suggested an increased probability of appearance as the distance to urban areas decreased, likely associated with the types of damaged summer crops, constituting 78.1% of the total. Summer crops like soybean, corn, and chili are primarily cultivated on small-scale reclaimed land and outdoor fields near water or residential areas, facilitating easy access for water deer inhabiting forest edges throughout Inje-gun.

Conversely, the limited number of autumn crops and concentration of croplands in specific areas contribute to the distinctive autumn distribution of water deer, with localized observed coordinates [51]. Despite this, cropland areas with a probability of water deer appearance of ≥0.6 were 17.11 km^2^ in summer and 16.44 km^2^ in autumn, suggesting an insignificant difference, likely due to the dependence level of water deer on croplands in autumn concerning food sources. While water deer exhibit access to multiple croplands in summer, their usage is relatively low due to the abundance of food sources close to their habitats. In autumn, with a reduced abundance, water deer dependence on croplands increases, leading them to access a small number of croplands more frequently.

The distribution probability of ixodid ticks associated with water deer was ≥0.6 in 198.50, 33.32, and 15.47 km^2^ for adult-, nymph-, and larva-stage ticks, respectively. This indicates a broad distribution for adult- and nymph-stage ticks, which are abundantly collected during summer, and a more confined distribution for larva-stage ticks, which are collected in abundance during autumn. This pattern is likely influenced by the predicted seasonal distribution of water deer in relation to cropland use.

In areas overlapping with croplands, the distribution was extensive for both adult and nymph stages at 74.95 km^2^, constituting 87.89% of the total area. In contrast, it was relatively narrow for nymph- and larva-stage ticks at 7.03 km^2^ (8.24%) and 6.00 km^2^ (7.04%), respectively. This outcome can be interpreted as a result of the local distribution observed in autumn, predicted for water deer based on weights assigned to the mean number of nymph- (271.6 ± 29.2) and larva-stage (1080.8 ± 82.6) ticks, which exceeded that for adult-stage (91.3 ± 13.0) ticks.

This research focused on water deer, designated as harmful wildlife, in cultivated areas near forests. Therefore, comprehending the habitat utilization of water deer in agricultural landscapes is challenging. Despite water deer being regarded as a species most frequently utilizing cultivated areas, there are limitations in not considering the dispersal of ticks from various host species, such as roe deer and raccoon dogs, entering cultivated areas [52]. Furthermore, there might be limitations in evaluating the extent to which local water deer act as dispersers of ticks, as the density of ticks inhabiting the area was not taken into account. Consequently, interpreting the dispersal patterns of various host species of ticks, including water deer, across agricultural landscapes nationwide based on the results of this study may prove challenging. However, the significance of this study lies in comprehending the previously unknown patterns of water deer in cultivated areas and the host utilization behavior of ticks, including the collection of ticks and larvae from water deer. Building upon this research, dispersion prediction within the study area was conducted. Considering this, it is crucial to undertake future studies that encompass cultivation area patterns of other host species and the tick index of parasitic ticks based on the results here. Expanding the research target areas according to agricultural types could contribute markedly to preventing tick-borne diseases for residents near cultivated areas in the country.

## 5. Conclusions

This study’s findings highlight the significant influence of host water deer on the abundance and dispersion of parasitic ixodid ticks, particularly in areas close to human residential areas and croplands. The distribution of ixodid ticks in croplands varies seasonally based on cropland use by water deer and the seasonal appearance characteristics of ixodid ticks at each life stage. As hosts for ticks, water deer are considered to serve as dispersers of ticks within their habitats, rather than playing a direct role in tick infestation or virus transmission to humans.

The observed three-fold-higher rate of SFTS infection in agricultural workers compared to the general population is likely associated with the dispersion of ixodid ticks in croplands facilitated by water deer, the primary accessing species of these areas, and the increased tick density in the surrounding regions. To mitigate human–water deer conflicts arising from overlapping habitats and prevent contact between ixodid ticks and humans, it is crucial to implement measures such as selecting crops that are less favored by water deer, based on food source analyses, and choosing suitable lands for reclamation via preliminary investigations of water deer density near croplands. These measures aim to reduce cropland dependency and usage by water deer while minimizing forest edges resulting from reclamation. Overall, these strategies not only decrease conflicts between humans and water deer but also serve to prevent interactions between ixodid ticks and humans.

## Figures and Tables

**Figure 1 animals-14-01153-f001:**
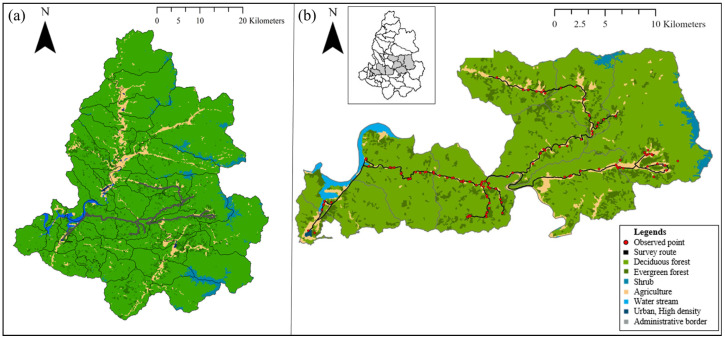
(**a**) Study site in Inje-gun showing survey routes. (**b**) Observed study site points with land cover types.

**Figure 2 animals-14-01153-f002:**
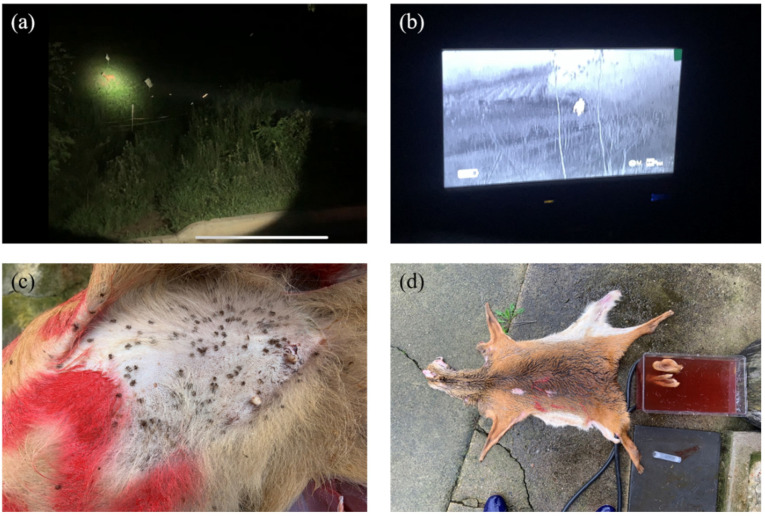
Water deer capture and ixodid tick collection. Searching for water deer in agricultural land using a (**a**) searchlight and (**b**) night vision camera; (**c**) collecting ixodid ticks from the water deer skin surface; (**d**) precipitating in water to collect ixodid ticks from the skin surface.

**Figure 3 animals-14-01153-f003:**
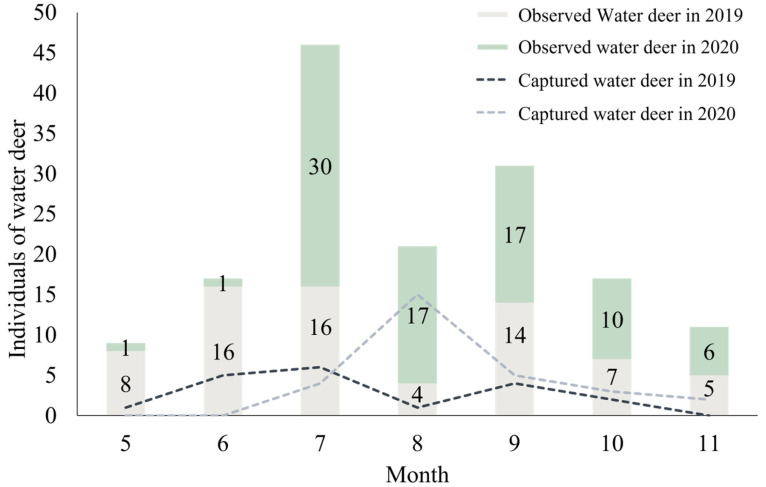
The number of water deer appearances throughout the study periods.

**Figure 4 animals-14-01153-f004:**
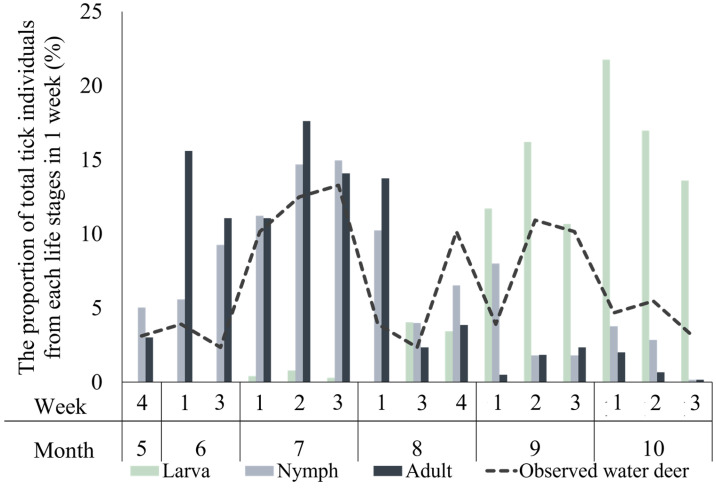
Weekly proportion of observed water deer and ixodid ticks at each life stage.

**Figure 5 animals-14-01153-f005:**
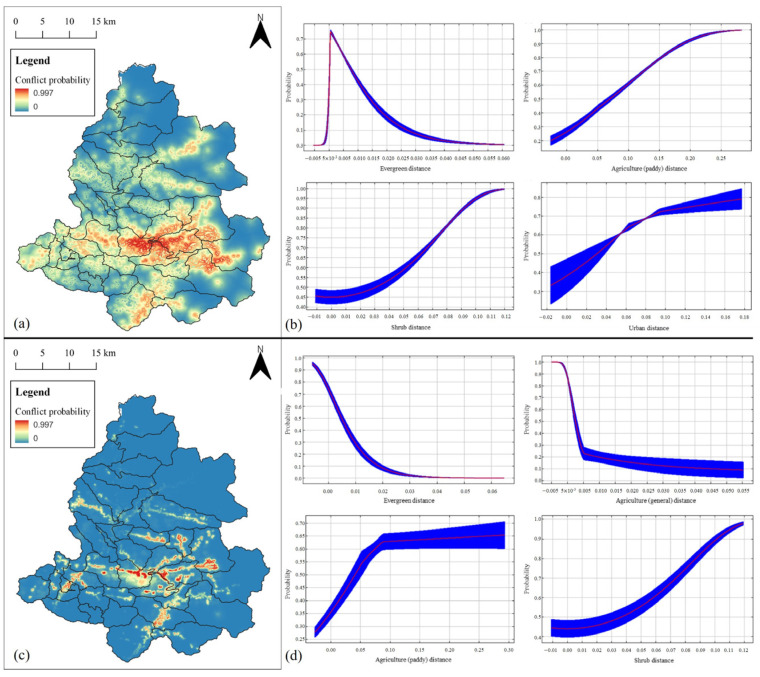
(**a**) Predictions of human–water deer conflict probabilities in Inje-gun during summer. (**b**) Response to probabilities in summer. (**c**) Predictions of human–water deer conflict probabilities in Inje-gun during fall. (**d**) Response to probabilities in fall. The red lines denote the mean values, while the blue background indicates the standard errors.

**Figure 6 animals-14-01153-f006:**
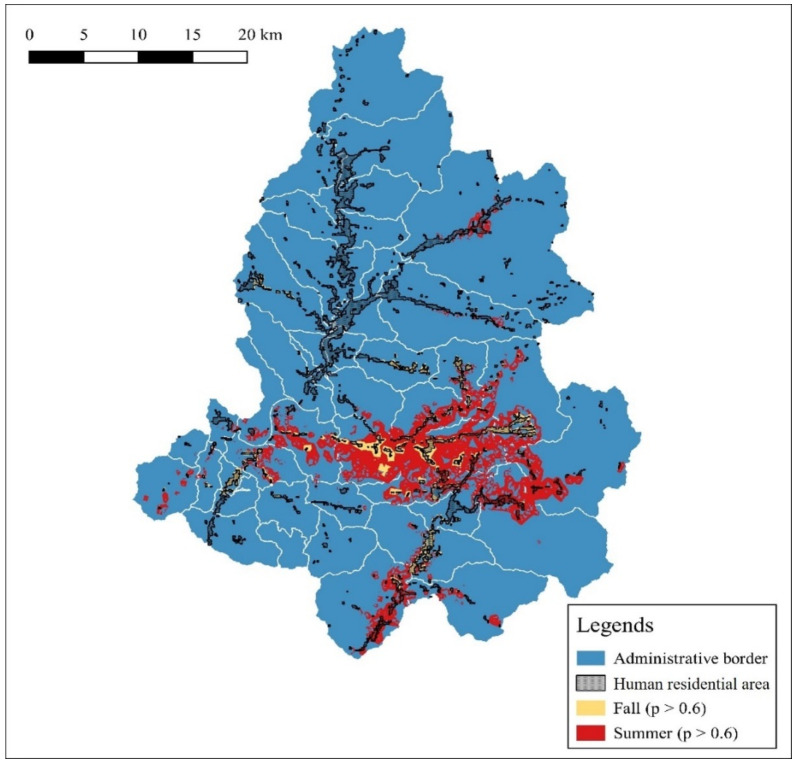
Overlap between human residential areas and water deer distribution (*p* > 0.6).

**Figure 7 animals-14-01153-f007:**
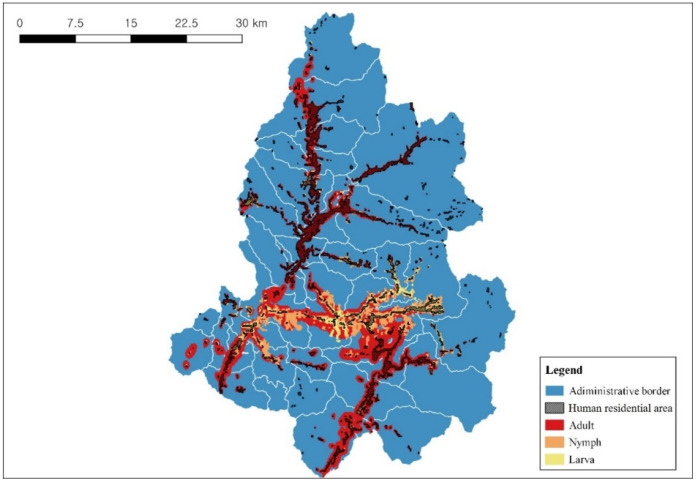
Probability prediction of ixodid ticks carried by water deer (*p* > 0.6) in Inje-gun.

**Figure 8 animals-14-01153-f008:**
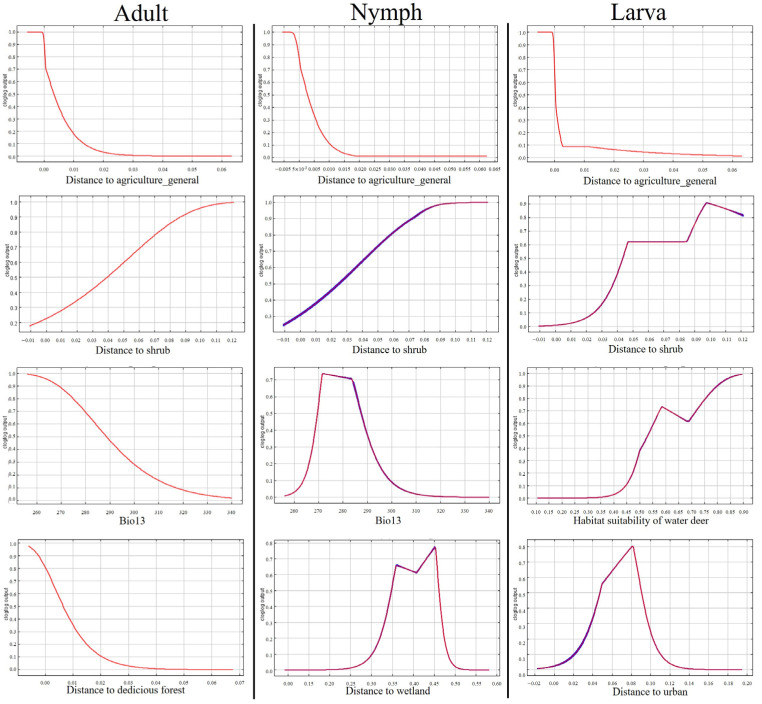
Response to probability across each life stage. The red lines denote the mean values, while the blue background indicates the standard errors.

**Figure 9 animals-14-01153-f009:**
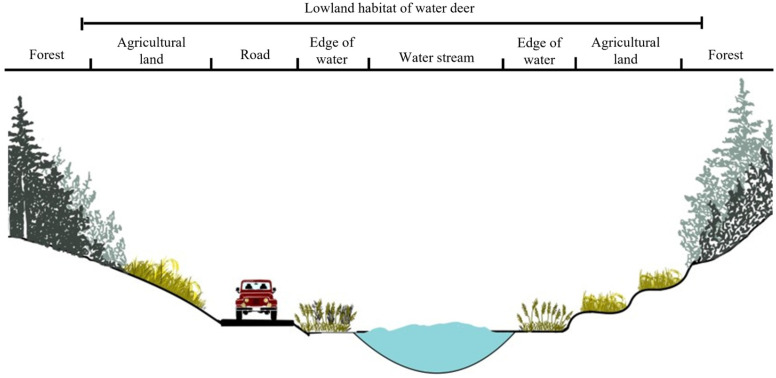
Lowland habitat of water deer around agricultural land in Inje-gun.

**Table 1 animals-14-01153-t001:** Variables used for modeling habitat suitability of water deer, human–water deer conflict zones, and ixodid tick probability in Inje-gun around cultivation areas.

Variable	Explanation
Bio1	Annual mean temperature
Bio2	Mean diurnal range (mean of monthly (max temp − min temp))
Bio3	Isothermality (Bio2)/(Bio7) × 100
Bio4	Temperature seasonality (standard deviation × 100)
Bio5	Maximum temperature of the warmest month
Bio6	Minimum temperature of the coldest month
Bio7	Annual temperature range (Bio5 − Bio6)
Bio8	Mean temperature of the wettest quarter
Bio9	Mean temperature of the driest quarter
Bio10	Mean temperature of the warmest quarter
Bio11	Mean temperature of the coldest quarter
Bio12	Annual precipitation
Bio13	Precipitation of the wettest month
Bio14	Precipitation of the driest month
Bio15	Precipitation seasonality (coefficient of variation)
Bio16	Precipitation of the wettest quarter
Bio17	Precipitation of the driest quarter
Bio18	Precipitation of the warmest quarter
Bio19	Precipitation of the coldest quarter
Elevation	Digital elevation model (DEM; GMTED provided by USGS)
Slope	Derived from the DEM using the Slope tool in ArcMap
Ruggedness	Derived from the DEM—standard deviation of the slope using focal statistics
Landcover	Extracted from World Land Cover 30 m (BaseVue 2013), Source: MDA information systems (MDAUS)
Deciduous forest (d)	Euclidean distance to deciduous forest land cover class (BaseVue 2013)
Evergreen forest (d)	Euclidean distance to evergreen forest land cover class (BaseVue 2013)
Shrub (d)	Euclidean distance to shrub (BaseVue 2013)
Grassland (d)	Euclidean distance to grassland (BaseVue 2013)
Agriculture_general (d)	Euclidean distance to general agricultural distance (BaseVue 2013)
Agriculture_paddy (d)	Euclidean distance to agricultural paddy distance (BaseVue 2013)
Wetland (d)	Euclidean distance to wetland (BaseVue 2013)
Water (d)	Euclidean distance to water (BaseVue 2013)
Urban (d)	Euclidean distance to an urban area (BaseVue 2013)

Note: All variables are continuous except land cover (categorical). (d) denotes the Euclidean distance (meter) to each variable. GMTED denotes Global Multi-resolution Terrain Elevation Data. USGS denotes U.S. Geological Survey.

**Table 2 animals-14-01153-t002:** Mean intensity of ixodid tick species and life stages (AF: adult female; AM: adult male; N: nymph) collected from water deer.

	*H. longicornis*			*H. flava*			*H. japonica*			*I. nipponensis*		
Month	AF	AM	N	AF	AM	N	AF	AM	N	AF	AM	N
5	8.0	7.0	89.0	1.0	2.0							3.0
6	44.2 ± 13.7	37.6 ± 8.5	127.4 ± 56.2		1.0	3.0 ±0.0					1.0	
7	70.5 ± 10.0	19.4 ± 3.9	267.8 ± 28.9	70.5 ± 10.0	19.4 ± 3.9	3.5 ± 0.6						1.5 ± 0.5
8	19.4 ± 9.3	7.2 ± 3.5	100.2 ± 24.9		6.2 ± 3.2	1.0	3.3 ± 0.9	4.0 ± 1.0	1.5 ± 0.5			4.0
9	2.0 ± 0.5		54.4 ± 16.1	4.0	7.5 ± 2.8	21.8 ± 5.9	4.0 ± 2.0	5.8 ± 4.1	2.0	1.0		6.3 ± 3.4
10			3.6 ± 1.0	1.0	5.2 ± 2.2	33.5 ± 6.9	1.0					20.5 ± 6.5

**Table 3 animals-14-01153-t003:** Number of captured water deer and mean intensity of each ixodid tick life stage.

		Month				
		6	7	8	9	10
Number of Captured Water Deer		5 (5)	10 (10)	16 (9)	9 (9)	5 (5)
MI of ixodid ticks	Larva	^abc^* 2.8 ± 0.7	^ab^ 47.6 ± 8.7	^a^ 278.8 ± 4.5	^ac^ 809.3 ± 235.9	^c^ 1080.8 ± 82.6
	Nymph	^a^ 128.6 ± 56.9	^b^ 271.6 ± 29.2	^a^ 101.1 ± 24.5	^a^ 44.7 ± 9.6	^a^ 38.6 ± 9.3
	Adult	^a^ 82.2 ± 21.1	^b^ 91.3 ± 13.0	^ac^ 24.6 ± 8.8	^c^ 8.4 ± 3.2	^ac^ 5.6 ± 2.1

* Notes: Different letters (^a, b, c^) indicate significant differences for each life stage within each month. On the contrary, it indicates that there is no significant difference between the number for the same letter in monthly values (*p* < 0.005). Numbers within parentheses indicate individuals carrying ixodid ticks each month. Data in both May and November are excluded due to the small sample size.

**Table 4 animals-14-01153-t004:** Spearman’s rank correlation coefficient (r_s_) among water deer and ixodid ticks at different life stages.

Season		Larva	Nymph	Adult
Summer	Water deer	** 0.819	* 0.609	0.138
	Larva		0.446	−0.144
	Nymph			* 0.549
Fall	Water deer	−0.065	* 0.583	0.187
	Larva		−0.469	0.014
	Nymph			0.297

Note: * *p* < 0.05, ** *p* < 0.01.

## Data Availability

The data presented in this study are available on request from the corresponding author.

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
