# Peer review of "Parasitic Characteristics of Ticks (Acari: Ixodidae) Collected from Water Deer (Hydropotes inermis argyropus) and Spatiotemporal Distribution Prediction within Host-Influenced Cultivated Areas"

_animals, 2024, doi:10.3390/ani14081153_

Round 1

Reviewer 1 Report

Comments and Suggestions for Authors

the review in the file below

Reviewer 2 Report

Comments and Suggestions for Authors

In this study, the authors examine the relationship between water deer use of habitat, the association of ticks with deer, and the potential risk of these interactions to humans in the habitats. This is achieved by capturing water deer in agricultural areas throughout the year, documenting the attached ticks, and measuring the associations between deer and different tick life stages in different seasons. The authors then use deer and tick data, combined with environmental variables, to model the probability of human-deer conflict and factors predicting the presence of deer and ticks. 

This is an interesting and comprehensive study and I enjoyed reading this and learning about the water deer use of this ecosystem and the association with ticks and potential tick borne disease risk driven by increased human-deer contacts. This research will be of interest both to those interested in wild deer populations, modelling to predict habitat suitability, and tick-borne disease, and will be an excellent addition to the Special Issue on Parasitic Arthropods of Vertebrates.

I only have minor concerns with this article - please find below some specific comments which will help authors to improve the clarity of Results and expand important areas of the Discussion.

Introduction 
line 57: change first 'their' in sentence to 'tick'

line 67: seems like an error in this sentence - delete "of the vector"

Methods
line 151: is this the correct reference [36]? It seems to be about trematodes.

Table 1: for the distance variables, what is the unit used? km?

Results
Figure 3 - this Figure is not discussed in the Results section and does not seem relevant to the results discussed. I suggest that this Figure might be better in Methods section as part (b) of Figure 1 to show where capture sites were. Spelling mistake in key - 'Administrative border'

line 193 - detail how many water deer were observed and of these how many were captured. It says 151 were "captured" but based on Figure 4, this is closer to number of observed.

line 195 - these Results are shown in Figure 4 not Figure 3.

line 212 - what about nymphs? They were also significantly high in July.

line 214 - it is not correct to say nymphs increased July - October. They increased from May to July, then decreased from August - October.

line 215 - adults appeared first in May, not June.

line 218 - indicate that Different letters indicate significant differences for each life stage between months (p < .005)

Figure 5 - it is not clear what "rate" refers to. Is it total number ticks or mean intensity?

Figure 6 - the labels used for each tick species are very confusing, because it is normal to label tick life stages with the letters L, N, and F (larvae, nymphs, female). I recommend that authors denote each tick species as follows: Hl, Hf, Hj, In for Haemaphysalis longicornis, Haemaphysalis flava, Haemaphysalis japonica, and Ixodes nipponensis, respectively. Also November could be deleted from the graph because it shows no data.

line 247-250: this could be described in more detail. Looks like correlation was significant for Larvae and nymphs but not adults. Combined tick counts for all life stages not shown in table but are described in text.

line 258 - distance to deciduous forest is described as an important factor, ut Figure 7b shows distance to evergreen forest.

line 262, 290, 297 - please detail what Bio7 and Bio13 are, i.e. annual temp range and precipitation of wettest month.

line 275 - consistent with the overall distribution of what? deer?

Figure 10 is labelled Figure 70.

Discussion
Figure 11 is labelled Figure 81.

line 401, 412, 416 - do the ticks truly overwinter "underground" or is it under vegetation mat layer?

Authors should include some discussion of the risk of SFTSV transmission to humans based on the study's results, including mentioning that H. longicornis is the major vector of this virus. Presumably the larvae do not play a big role in virus transmission, so June - August represent the highest risk for humans to contract tick-borne diseases in these agricultural areas. Does this match with the seasonality of human SFTS cases in the region?

Line 489 - are exclusion strategies such as deer fencing also options to mitigate risks of deer-human overlap in these areas?
